# Molecule Sensitive Optical Imaging and Monitoring Techniques—A Review of Applications in Micro-Process Engineering

**DOI:** 10.3390/mi11040353

**Published:** 2020-03-28

**Authors:** Marcel Nachtmann, Julian Deuerling, Matthias Rädle

**Affiliations:** Reseach Center CeMOS, Mannheim University of Applied Sciences, Paul-Wittsack-Str. 10, 68163 Mannheim, Germany; m.nachtmann@hs-mannheim.de (M.N.); j.deuerling@hs-mannheim.de (J.D.)

**Keywords:** surface scanning optics, Raman, near infrared, middle infrared imaging, scanning, multimodal spectroscopy, local reaction control techniques, microchannel

## Abstract

This paper provides an overview of how molecule-sensitive, spatially-resolved technologies can be applied for monitoring and measuring in microchannels. The principles of elastic light scattering, fluorescence, near-infrared, mid-infrared, and Raman imaging, as well as combination techniques, are briefly presented, and their advantages and disadvantages are explained. With optical methods, images can be acquired both scanning and simultaneously as a complete image. Scanning technologies require more acquisition time, and fast moving processes are not easily observable. On the other hand, molecular selectivity is very high, especially in Raman and mid-infrared (MIR) scanning. For near-infrared (NIR) images, the entire measuring range can be simultaneously recorded with indium gallium arsenide (InGaAs) cameras. However, in this wavelength range, water is the dominant molecule, so it is sometimes necessary to use complex learning algorithms that increase the preparation effort before the actual measurement. These technologies excite molecular vibrations in a variety of ways, making these methods suitable for specific products. Besides measurements of the fluid composition, technologies for particle detection are of additional importance. With scattered light techniques and evaluation according to the Mie theory, particles in the range of 0.2–1 µm can be detected, and fast growth processes can be observed. Local multispectral measurements can also be carried out with fiber optic-coupled systems through small probe heads of approximately 1 mm diameter.

## 1. Introduction

In this article, we focus on remote control techniques that may or may not yet be widespread. Most of the presented measurements were carried out at the Center for Mass Spectrometry and Optical Spectroscopy—CeMOS, an interfaculty institution of the University of Applied Sciences in Mannheim, Germany.

Optical time and space resolved measuring technologies in the UV- and visible range are used for the better understanding of flows, mixing processes, and the control of reactions in microchannels. Common image analysis results in two-dimensional images that are measured in reflecting, or if the microchannel has been specially adapted and manufactured, transmitting arrangements. For transmitted light, both the bottom and the lid of the microchannel must be transparent. 

Obtaining an increase of contrast is possible in several ways, e.g., by restricting the depth of field from classic image analysis. Only a narrow, defined detection plane results in a sharp image. In transmission, the obtained concentration values are the mean values for the respective vertical axis intercept. Complex tomographic 3D-scanning instruments suppress this effect but require more time to capture the necessarily high number of images. 

Classical methods of microscopy and image analysis allow for insights into the spatial distribution of gases and fluids, as well as their temporal development. Contrasting the moving phases greatly improves the optical contrast. Fluorescence marking is common in this context, as it requires low concentrations of markers. The main disadvantage of this technique is the influence of the marker on fluidics and the lack of selectivity. Self-fluorescence detection is only possible in very few cases. These techniques cannot detect the local, time-resolved, concentration-controlled identification of molecular species and a possible tracking of mixtures, inhomogeneity, reactions or deviations in reaction behavior, nor the occurrence of by-products. Therefore, fluid science is in need of measuring systems that combine a high spatial and temporal resolution with molecular selectivity and no disturbance of fluidics. For optical technologies, the measurement range is, depending on the detector used, not limited by the range of perception of the human eye. All known approaches have advantages and disadvantages, and the current state-of-the-art approaches are explained in following article.

## 2. Materials and Methods 

In this paper, an overview of how optical measurement techniques can be used to quantify effects in microchannels is given. To illustrate this, the interaction of light with a specific target or matter in general is explained. Light, in this case, extends from the ultraviolet range via the visually visible range via near-infrared to mid-infrared and should not—as in common speech—be limited to the visible range only. 

Light interacts with the product, both elastically and inelastically. Elastic light scattering means that the incoming photon does not cause a molecular change of the energy states and leaves the medium with the same wavelength. Inelastic light scattering means that vibrations or rotations of the molecules, as well as the lattice vibration phonons of the solid bodies, are excited [1,2]. Electronically excited states can also be achieved, in particular by irradiating with ultraviolet light [3].

First, it is necessary to note which basic beam paths geometries, like transmission, backscattering, or remission into other angles, can be applied for the measurement techniques to be described. Figure 1 headlights the different types of possible interactions between light and a particle.

Based on the different interaction types, a variety of measurement techniques are possible. One of the most commonly used techniques is the transmission setup. A change of the outgoing light in transmission can take place through the absorption of certain wavelengths by particles, other disperse substances such as drops, aerosols, or through the molecular absorption of fluids or gases [2,5]. 

In fluids, the dissolved substances are often referred to as molecularly-disperse in order to distinguish them from macroscopically disperse substances like particles [6]. For solutions, i.e., molecularly-disperse dissolved substances, transmission is one of the most frequently used application areas [3]. Absorbed light can be described with the Lambert–Beer law according to the following formula in which E stands for extinction or absorption, I_0_ is the emitted, I is the transmitted light, ε is the extinction coefficient, c is molar concentration, and d is the layer thickness: [3]
E_λ_ = log_10_ (I_0_/I) = ε ∙ c ∙ d(1)

The decisive factors here are the molar concentration c and the extinction coefficient ε of the irradiated wavelength. Usually, the experimental setup defines the layer thickness d. For disperse materials, leaving the primary optical axis for measuring the absorption in combination with scattering is possible. 

Backscattering, side scattering, and forward scattering are common angular possibilities [3]. The basic theory for disperse materials is accumulated in almost all literature in the application of the Mie scattered light theory and simplifications of the range of small particles or large particles. In the range of small particles, the Rayleigh approximation is often used, as it is applicable when the light is smaller than about a quarter of the irradiated wavelength. For large particles, Fraunhofer diffraction is commonly used. Here, the particles have to be larger than about five times the wavelength of the exciting light. All these theories imply that the particles are round spheres. Theories for non-round particles are almost unheard of [7,8,9,10,11,12]. 

By using transmission, the extinction coefficient can be determined from the intensity loss, which includes the phenomena of absorption and scattering [3]. Applied to the geometries of microchannels, a measurement setup based on transmission has the decisive disadvantage of the need to be able to illuminate through the microchannel. This requires two transparent side surfaces. Some applications may need temperature control. With this setup, temperature control is usually more problematic than using non-transparent metallic boundaries. In transparent designs, the heat transfer coefficient is a further limitation. Therefore, the pure transmission arrangement is usually not suitable for microchannel applications.

Another geometry is the so-called attenuated total reflection technique (ATR). A light beam, irradiated laterally at a certain angle, is reflected at the boundary between the crystal or optical material of the sensor and the fluid. With the corresponding theory, the quantum-mechanically explainable evanescent wave penetrates a few wavelengths into the medium and is potentially absorbed there. The ATR technique is relevant when high concentrations and high extinction coefficients simultaneously occur. This effect can be found in the ultraviolet range below the wavelength of 320 nm, i.e., UV-B or UV-C systems. With dyes, the effect also take place in the visible range. Mostly, the ATR setup is used in the mid-infrared range, because the excitation of the ground states of vibration at this point is accompanied by high extinction coefficients [3].

Light scattering measurements are used for dispersed-phased products or if a change in the disperse phase is a suitable control or quality parameter for such processes. Precipitation reactions, for instance, with particles precipitating from a molecularly-dissolved starting material that reaches supersaturation or grows in a particular way such that scattering is increased, are also suitable for scattering measurements. Optical scattering correlates in the same way as transmission via the Mie scattering theory or via Rayleigh or Fraunhofer diffraction by using the same mathematical equations. The scattered light measurements can be separated into geometrical subdomains. A decisive technological question and boundary condition is the arrangement regarding the angle of the scattered light. In a laboratory environment, 90° scattering is common, but this scattering angle is difficult to adapt when applied to microchannels [7,8,9,10]. Target process variables are mostly local concentrations. In microchannels, for example, local particle concentrations and the velocities of particle size and concentration changes are relevant variables. The determination of kinetic parameters using these process variables is possible. 

### 2.1. Particle Detection

The relevant particle size for many products is about one micron. For theoretical description, the Mie theory, postulated by Gustav Mie, suits best. Currently, several programs are available for the calculation of diffraction patterns; these include the free-to-use algorithm by Wiscombe et al. [12]. An example for such diffraction patterns is the angle dependence of scatter light intensities. Mostly, a logarithm scale best suits the matching of a wide illumination distribution for different angles. This requires special detectors to suit the whole intensity distribution. For different particle sizes, the angle dependence, which is a special boundary condition in Mie theory, differs. Mie back scattering is shown for 180° in Figure 2 [7,10].

In addition, the quotient of the wavelength and particle size correlate with angle dependence. This means that the particle growth can be determined by using several different wavelengths in the visible area. Usually, two wavelengths are sufficient. Different wavelength bands can be obtained by using different light-emitting diodes (LEDs) instead of a halogen lamp or by using different optical filters in front of the detector [11]. Broadband detectors accumulate all irradiated light. By using different LEDs, simultaneous measurements are, out of the box, not possible. Optical filters are needed or the LEDs have to be used in an alternating fashion. The oscillating intensity quotient, obtained from those two wavelengths, correlates with the actual particle size and matches the theoretical calculations following Mie, as shown in Figure 3 and Figure 4. 

Several limitations may apply, but particle size can, in most cases, be determined with an acquisition rate of over 1000 Hz. Limitations may be the particle size itself. Only in the Mie area is the quotient oscillating. If particles are smaller 100 nm, this effect cannot be observed. The presented technique is also not applicable for particles in the Rayleigh area. Growing monodisperse particles can be measured up to about 2 µm. For bigger particles, the oscillation is too frequent to be separated for each particle. Mistakes and mix-ups presumably increase with ascending particle size. This method may be limited to a specific particle size range, but it performs well with small component size and the measurement frequency [8,10,11]. Furthermore, probes for scattered-light measurements can be extremely small. For example, manufacturing backscatter probes even in a blunt cannula with an inner diameter of 1.1 mm is possible [13].

In such a blunt cannula (Figure 5), several glass fibers can be included for different light sources and detectors. This technology is used for measurements in micro channels to obtain particle size, concentration, and growth [13].

The method is certainly special and is limited to the specified particle size range, but, compared to all other measuring methods, it has the huge advantage of being small, i.e., miniaturized, and of high-speed detection. For the questions posed here, scattering at an angle of 180°, i.e., backscattering, is often a more suitable arrangement. Additionally, 45° backscattering— or 135° from the point of view of angle geometry—is a popular arrangement. The 135° scattering angle has particularly established itself in the field of color pigment monitoring [8,10,11]. The entire optical arrangement of the emitter and the detector can be realized in one stable, steady instrument as an easy-to-use setup with a small physical gap between the transmitter and the receiver. This leads to reduced surface contamination effects in the measurement signal—bypassing the Tyndall effect (Figure 6).

In addition to elastic light scattering, spectroscopic methods like molecular excitations are relevant, especially for the quantification of molecule concentrations. Here, the whole range of spectral technologies is available and is explained in the following sections with regard to their theoretical explanations and these technologies’ possible applications.

### 2.2. Ultraviolet/Visible (UV/VIS) Spectroscopy 

Starting with UV spectroscopy, the electrons are usually excited into higher orbitals by so-called electronic excitations. These are usually characterized by high extinction coefficients and have very wide bands or absorption edges. Therefore, UV spectroscopy has a high sensitivity but a low selectivity. Real process control is only possible in exceptional cases where the molecules make this possible [3,14]. Examples can be found in the conversion of nitrobenzene and sulfuric acid to nitrobenzene sulfonic acid. In this process, the nitrobenzene band disappears, and, thus, distinguishing nitrobenzene concentrations from the beginning to the end of the process is easily possible. In VIS spectroscopy, the same effects occur as in UV spectroscopy, though only for substances that are usually recognizable as colors in the visible spectrum. Therefore, the effects occur with dyes (molecularly-disperse dissolved dyes) or with color pigments (particulate substances containing chromophores) [4].

### 2.3. NIR-Spectroscopy

Near-infrared (NIR) spectroscopy uses the fact that almost all fluids in the NIR range provide absorption bands. Thus, they are usually distinguishable. In the world of chemical monitoring, these methods are highly established because it is easily possible to guide the light via glass fibers to chemical reactors [3,14,15]. Miniaturized fiber optic probes are also suitable for microchannels [4]. The excitation of the second or higher harmonics of the vibrational–rotational transitions of molecules caused by the photons is used. The fundamental oscillation of the same molecules is in the mid-infrared range [3,14,15]. The extinction is relatively weak and only measurable at higher concentrations (above approx. 0.1% in fluids) [4].

The photons used in near infrared spectroscopy usually excite the harmonics of present molecules. Furthermore, excitations can occur through combination oscillations while supplying the necessary energies. The incoming photon must therefore trigger an OH oscillation in the water molecule and additionally excite an oscillation in the bending angle of the water molecule via the so-called banding mode. The water molecule, in particular, is very dominant in the near infrared range. The coupling of the electromagnetic wave of the photon is a dipole interaction, so molecules with a strong dipole moment have high extinction coefficients in the near infrared range [16].

Due to the large number of possible combinations of oscillations and rotations, the near-infrared spectra are complex. For the pictorial measurements discussed in this article, two possible methods are convenient: a scanning method, which records a complete spectrum per point, or photometric methods, where only one wavelength is recorded at a time but can be displayed over a large area by using infrared cameras. Both methods have advantages and disadvantages. An advantage of the spectroscopic method is that a wavelength resolution of the laboratory apparatuses that measure the samples. The whole range of known reaction monitoring is available. A disadvantage is the time required for the measurement. About 1 s is required per measuring point, so a flat image takes several hours to measure. An alternative are area measurements. Broadband photo filters or light-emitting diodes in the near-infrared range on the transmitter side are necessary. The advantage here is that an image is quickly acquired (about 40 ms), but only one wavelength is recorded. The interpretability of the results is therefore often limited because the installed filters or light sources do not get close to reaching the wavelength resolution of the spectrometers. For this type of application, we later focus on mixing processes.

### 2.4. Mid-Infrared Spectroscopy

The mid-infrared region is characterized by the excitation of the ground states of vibration; the theory is sufficiently well-known in the literature. The significant advantage of mid-infrared spectroscopy is its selectivity in combination with its high sensitivity with regard to molecules and their detection limits on the concentration scale [3,16].

With this type of spectroscopy, fundamental oscillations of the molecules are scanned with, in contrast to NIR spectroscopy, much higher excitation cross sections. The advantage of this is the possibility of detecting even small concentrations down to the parts per million range. Due to the sharper bands, the selectivity towards NIR is also greatly increased. A disadvantage is the complex technology required, which means that a robust design of the device is difficult to realize. Existing fiber technologies are unstable, have limited spectral range, and have no long-term stability in harsh environments. The extremely high absorption of water also proves to be disadvantageous. Water-containing substance systems are superimposed by water absorbance, and other substances—especially with low concentrations—are only found with difficulty [3,16].

### 2.5. Fluorescence Spectroscopy

In fluorescence spectroscopy, the incident light excites short-lived electronic states within the molecule. After typical times in the nanosecond range, the molecules return to their ground state. Often, however, they do not fall back to the same oscillation state of the electronic ground state, instead falling to excited oscillation levels. For this reason, the wavelength emitted is often longer, and the photon energy is thus lower than that of the incident light. The fluorescence is extremely sensitive to detection. However, it requires a molecule-specific electronic state. Therefore, only a part of the molecule fluoresces, and imaging is limited with regard to the selection of molecules [4,14,17].

### 2.6. Raman Spectroscopy

Raman technology is the strongest upcoming technique at the moment. It scans the fundamental oscillation of molecules, but, in contrast to mid-infrared (MIR) spectroscopy, it does so by exploiting the polarizability of molecules. Stimulated with photons in the visible range, the molecules are short-time excited into a virtual state and fall back instantaneously into another vibrational–rotational state of the electronic basic level. This has the advantages that, depending on the used excitation wavelength, classical fiber-optic sensors are applicable for Raman measurement techniques and the high selectivity of MIR spectroscopy is present. On the other hand, the Raman effect is extremely weak. Strong lasers used as excitation sources and long integration times are necessary. Technological advancements in the last few years have had a strong and positive impact on this situation [3,15,18,19,20]. 

Table 1 shows a comparison of all mentioned spectroscopic methods for better comparison, highlighting the major advantages and disadvantages of each technique.

## 3. Results

In this next part, a different application, based on the results, is presented.

### 3.1. NIR Image Analysis

Analogous to image analysis in the UV/VIS range, charge-coupled device (CCD) cameras are used, and wavelength selectivity with optical filters is utilized either on the emitter or on the detection side. Cameras on silicon-based chips have a limited spectral range and are not sensitive in the NIR wavelength region. Instead, cameras based on indium gallium arsenide (InGaAs) are used, and these are sensitive in the 900–1600 nm range. Exact wavelength ranges may vary depending on the exact InGaAs detector used. In combination with NIR LEDs, acquiring images with wavelength-selected brightness levels according to the absorption bands of fluids for locally-fluctuating concentrations is possible. Acquiring reference images without the presence of any absorbing substances is important. The computer-based optimization of images, like anti-shading methods, can be used to compensate for inhomogeneous illumination problems. Advantage of NIR LED illumination as an alternative to broadband illumination and the use of sequentially used filters include the simplicity of its design and its fast change of wavelengths. Without special equipment, image sequences of 25 full frames/s are achievable, and, thus, moderately fast changing processes are accessible for detection. With individual wavelengths, pulses down to microseconds can also be captured [21]. By mathematically linking the remission at the surface with the absorption of the investigated fluids, extinction data and the resulting layer thickness distributions can be visualized. In Figure 7, you can see the measurement data of water strands.

A uniform illumination of the examined surface is important for later measurements, especially with curved surfaces (e.g., pipes). Here, the refractive index difference to the gas space causes light deflection, which can lead to the significant misinterpretation of the measurement signals. In the present case, the deflections were already significantly suppressed. This reduction is possible by distributing the light directions via a special dome illumination. The work so far has concentrated on the application of film thickness measurements. In addition, reactive processes, where both the fluid concentration and the film thickness distribution changes, are to be measured [21].

### 3.2. MIR Image Analysis

In the field of thermal imaging, mid-infrared (MIR) image analysis is common. However, depth of field and contrast are usually not sufficient for scientific purposes. The method of MIR scanning has proven to be more favorable. Commercial devices typically reach scanning velocities of one-to-two measuring points per second. Therefore, the acquisition of images is very time-consuming. A two-stage procedure for data acquisition proves to be more capable. The first stage, image scanning with full spectrum width, is selected for each pixel. Here, the assignment of interesting statements to the suitable wavelengths can be assigned via gold standards for narrowly-defined local conditions [22]. In the following step, the interesting wavelengths and target cutout are selected and monitored with up to 300,000 measurements per second with a spatial resolution of 20 μm in the fast scanning mode using quantum cascade lasers for routine sample measurements [23]. 

The special confocal beam path provides a tolerance towards distance variations to the target. As proven, a fast, flat, and confocal absorption measurement in the middle infrared range, from 3 to 5 μm, is possible (Figure 8). With these techniques, unequal surfaces with topographies can be sampled, and coatings with a layer thickness of less than one micrometer are eligible for molecule selective detection. It is possible to transfer the measuring principle to further applications [23]. 

### 3.3. Raman Scanning Image Analysis

As explained above, Raman spectroscopy requires high laser power and long exposure times. With precisely focused and installed Raman probe-heads (Figure 9) 2D Raman scanning is possible [24]. Even 3D Raman scanning seems to be achievable.

In the measurements conducted so far, this method has made it possible to visualize 2D Raman scans in fluids for the molecular quantification of the concentration profiles in microchannels and to perform the fundamental investigation of the mixing processes of different accessible fluids, as shown in Figure 10 [24].

The measurement rate correlates with the spectral resolution. Increasing the measurement rate decreases the possible resolution. Further advancement towards higher measurement speeds is possible by limiting the numbers of detected Raman shifts by replacing classical Raman spectroscopy by Raman multichannel photometry [18,24]. Large-area detectors promote Raman photometry by replacing the necessary dispersion and thus small-area detections in the spectrometer with filter techniques and single photon counters (such as a photomultiplier [25]). The area and sensitivity gain lead to a considerable increase in measurement speed. Spectral information is lost, but this can be of minor importance for known materials [18,26].

Multichannel-Raman photometry shines for process control with rather simple matrices. In Figure 11, the successful reaction tracing of binding CO_2_ to an amine until saturation is plotted, with a comparison of spectroscopic and photometric measurements [18]. The absorption of CO_2_ is well-known in the literature [27]. In photometers, faster detection rates, due to single photon counting detectors, are available and are subsequently able to resolve the reaction progress in a more detailed fashion. Both the spectrometer and photometer are able to trace the reaction progress. Complex matrices demand an increasing number of measurement channels. The measurement speed remains, but at least the financial advantages fade over the number of channels implemented compared to a spectrometer system. The sweet spot for Raman photometry, combining fast measurements and cost reduction, is between two and four channels. Raman photometers use an excitation laser as a light source. Every measurement channel is equipped with at least one small-band optical filter and one single-photon counter. A beam-splitter has to be added for a two-channel-photometer, and three beam-splitters for a 4-channel-photometer if an equal signal distribution is favored [18].

### 3.4. Scattering Light Measurement Technology

Scattering light techniques (elastic light scattering) can be used to control the formation of disperse phases, e.g., due to product failures. Fiber optic backscattering was used with a detection rate of several full spectra per second [9]. The fast detection rate is mandatory to visualize the precipitation of inorganic products in the millisecond range. Microchannels, in particular, allow for strong concentration gradients and, thus, pH or temperature gradients. This can lead to extremely fast precipitation processes. Fast measurement technologies are required. In geometries with constant flow and events occurring constantly at each location, kinetic fields for precipitation processes can be developed in combination with fast backscatter measuring systems with scanning equipment [4].

### 3.5. Fluorescence Techniques

A method for malignant tissue detection is a laser-induced fluorescence, noncontact-imaging approach, as shown in Figure 12. 

The malignant tissue is labeled with a fluorescent marker. The selected marker can be excited in the first optical window of the tissue. The optical window is defined as a wavelength region between the absorption band of hemoglobin (Hb) and water. The emission also has to be in the first optical window to avoid absorption. This method is not supposed to replace common technologies, like computer tomography (CT) or magnetic resonance imaging (MRI), but it is meant to supplement the highlighting of the malignant areas to reduce time that is needed with common technologies. The advantages of this method, compared to common technologies, are its greatly increased measurement rate and low price [28].

### 3.6. Combination Techniques: UV/VIS/NIR/Fluorescence

This example illustrates how the different modes described can be built into one device. This allows for, e.g., the distribution of lipids via Raman scanning. The additional information of the local moisture that is gathered from NIR scanning is helpful for gathering information on wound healing and also for reaction monitoring in the microchannel. An optimal control technology seems to be the combination of different techniques and the selective wavelength information generated by different wavelength regimes. The combination provides a flexible application for divers scientific problems.

The backscattering needle probe was designed as a setup for simultaneously, multimodal spectroscopic measurements. The basic setup, presented in Figure 13, contains glass fibers for ultraviolet, visible, near infrared and fluorescence spectroscopy. 

The probe head consists of a name-giving needle and seven bare glass fibers for illumination and detection [13]. Glass fibers can be used to transport light from the probe head to the spectrometer or detection unit [29,30]. The real time detection of malignant tissue in in vivo measurements is possible. A back cut can be applied for better tissue penetration. Measurements can be done with either a spectrometric or a photometric set-up. For evaluation purposes, the measurements can be done with a spectrometer, and a virtual photometer can be calculated. This approach has the advantages that it has been used for proof of concept photometric measurements and reference measurements can be simultaneously obtained. In addition, the received data are compressed. The monitoring of several metabolic parameters, like Hb, deoxy Hb, scattering, fat, and auto fluorescence, is possible. Clustered in groups for malignant, marginal, and healthy tissue, the obtained data are analyzed by a 2D principle component analysis (PCA). All samples can be assigned to their respective groups. Classical histology validates all measurements [13].

## 4. Discussion

In summary, the processes in microchannels, focused on the local concentration distribution of molecules, can be isolated and processed when the target molecules are spectroscopically accessible. In currently available technology, this usually means that molecules can be excited to vibrate. This applies, for example, to water, hydrocarbons, nitrates, and phosphates but not to the dissolved portion of hydrochloric acid, hydrogen fluoride, hydrogen bromide, or other ion concentrations without covalent bonds.

In principle, two classes of devices can be distinguished with regard to pictorial representations:Scanning systems.Simultaneous imaging systems.

Scanning systems are generally slower. Therefore, current technological development is focused on reducing the necessary number of wavelengths and increasing their detection sensitivity. These two measures promote an increase in throughput. For known molecules and non-complex matrices, the spectral resolution of spectrometers can be traded for faster measurement rates in photometers.

The current level is approx. 300,000 measuring points/ss for MIR, NIR, and UV/VIS, and it is possibly 1000 measuring points/ss in relation to Raman measurement technology [23]. 

The differences between the scanning technologies result from their respective applications and boundary conditions. In the presence of water, the strong water absorption in the MIR and NIR often covers the more interesting spectra of the searched molecules that may be present in smaller concentrations. Here, Raman technology proves to be beneficial, since the dipole character of the H_2_O molecule has limited relevance. On the other hand, Raman spectra show low extinctions, excluding thin layers and very low concentrations. 

Molecule selectivity increases from shorter to longer wavelengths: UV < VIS < NIR < MIR. However, with longer wavelengths and the associated higher molecular selectivity, technological effort also increases. The selectivity of Raman spectroscopy is similar to that of MIR measurements: comparatively high. 

With simultaneous imaging flat camera systems, all pixels are simultaneously recorded during a measurement. The advantage of simultaneity is often accompanied by a loss of wavelength selectivity and, thus, molecular selectivity. Because only one wavelength can be recorded at a time, the filter or selective illumination is, because of technical limitations, always somehow broadband. In the case of the necessary planar illumination, compromises usually have to be made with regard to detection optics, which leads to a loss of image quality. Contrast reduction, shading, and image distortion are the visible effects of this process.

Ultimately, the decision for a suitable measuring system is made based on the application and the resulting boundary conditions.

## 5. Patents

Patent Nr. DE102018105067A1: Bildgebendes System zur nichtinvasiven optischen Untersuchung von Gewebe in der Tiefe; Ahlers, Rolf, Prof. Dr., 64625, Bensheim, DE; Braun, Frank, 69226, Nußloch, DE; Hien, Andreas, 68161, Mannheim, DE; Rädle, Matthias, Prof. Dr., 67273, Weisenheim am Berg, DE.

Patent Nr. DE102014107342A1: Vorrichtung und Verfahren zur Erkennung von Krebstumoren und anderen Gewebeveränderungen; Braun, Frank, 69226, Nußloch, DE; Gretz, Norbert, Prof. Dr., 68259, Mannheim, DE; Rädle, Matthias, Prof. Dr., 67273, Weisenheim am Berg, DE.

## Figures and Tables

**Figure 1 micromachines-11-00353-f001:**
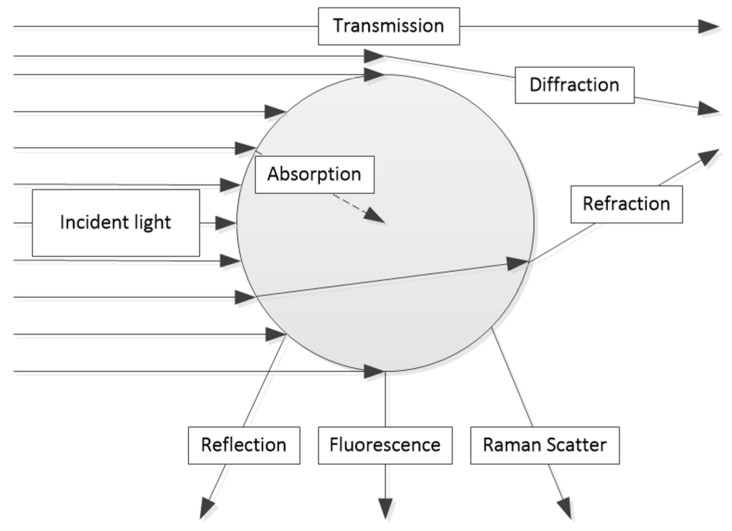
Types of interaction between light and a particle [4].

**Figure 2 micromachines-11-00353-f002:**
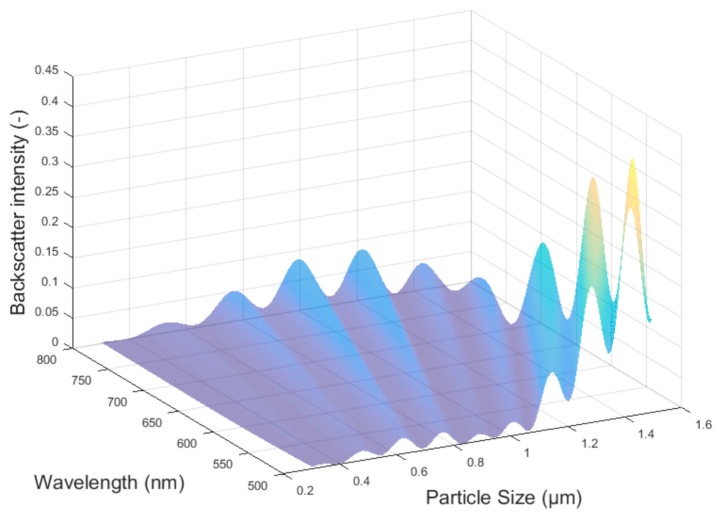
Mie back scattering intensity at 180° vs. particle size and wavelength [7].

**Figure 3 micromachines-11-00353-f003:**
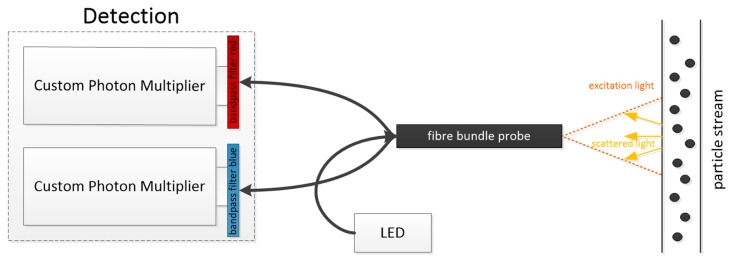
Detection setup with different bandpass filters [4].

**Figure 4 micromachines-11-00353-f004:**
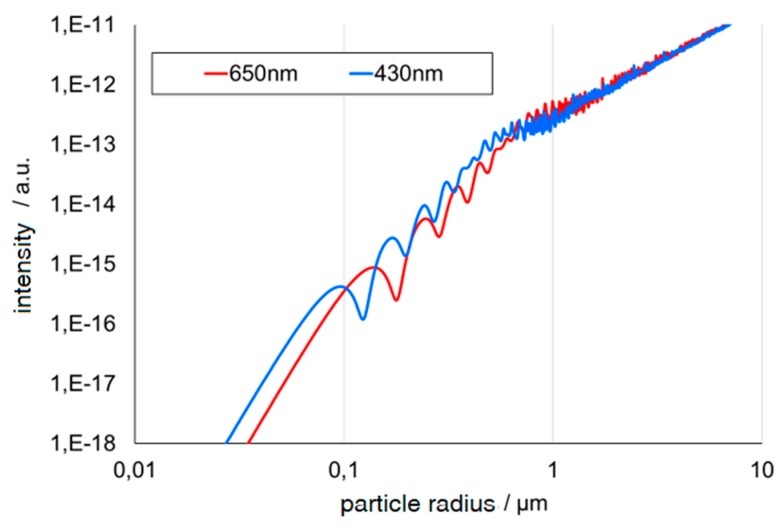
Oscillating intensity from two wavelengths [4].

**Figure 5 micromachines-11-00353-f005:**
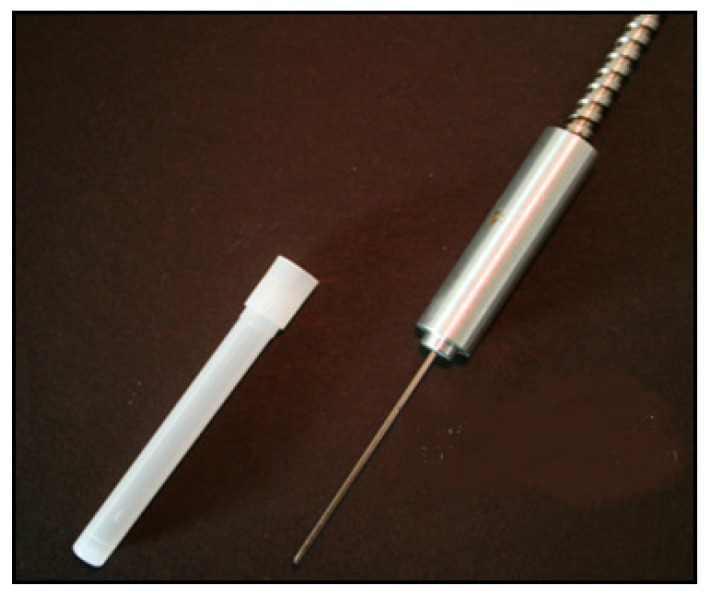
Cannula sterican 19G, 30° bevel [4].

**Figure 6 micromachines-11-00353-f006:**
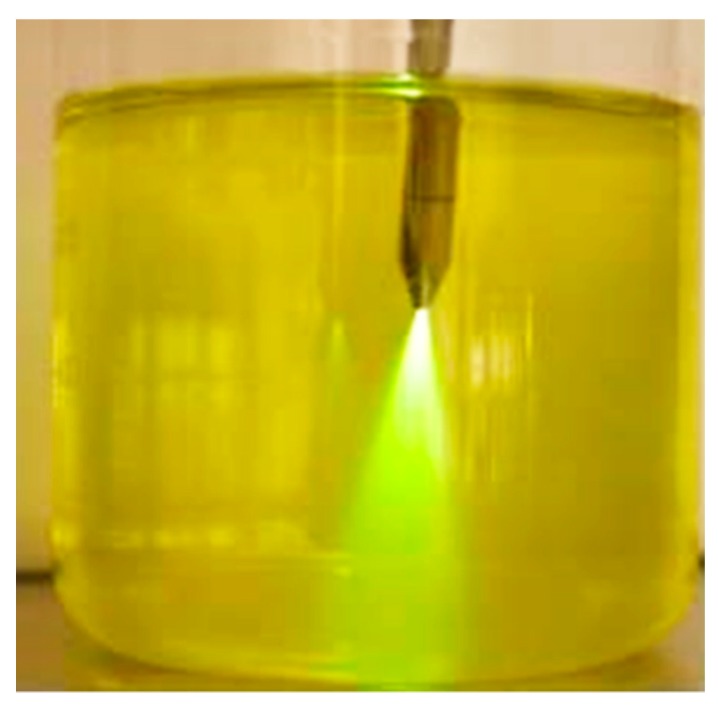
Tyndall effect, 10 ppm fluorescein sodium in water [4].

**Figure 7 micromachines-11-00353-f007:**
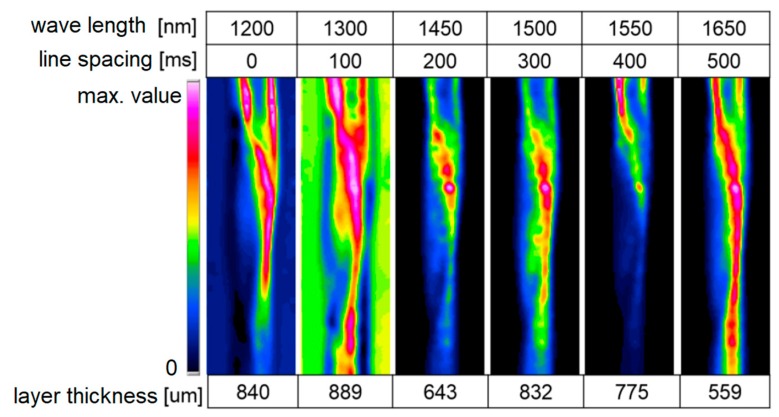
Near-infrared (NIR) measurement on water strands at all different wavelengths 274 × 142 mm (220 × 220 DPI), water glycerin [21].

**Figure 8 micromachines-11-00353-f008:**
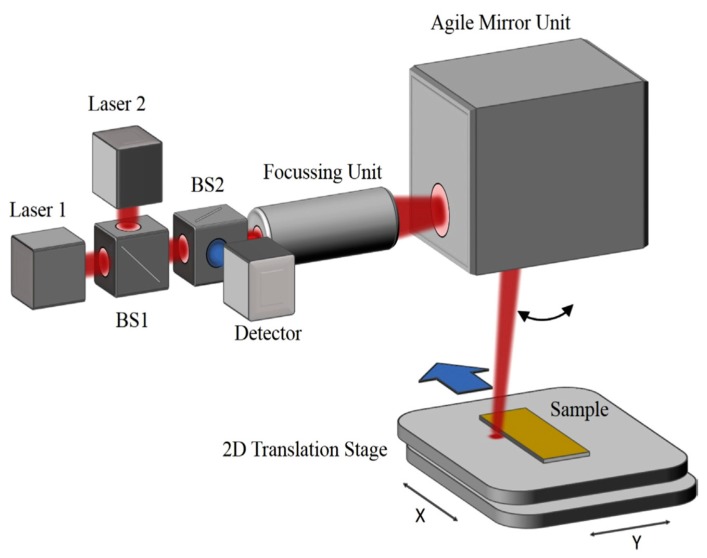
Schematic view of the mid-infrared (MIR) scanner setup with confocal mounted detector [23].

**Figure 9 micromachines-11-00353-f009:**
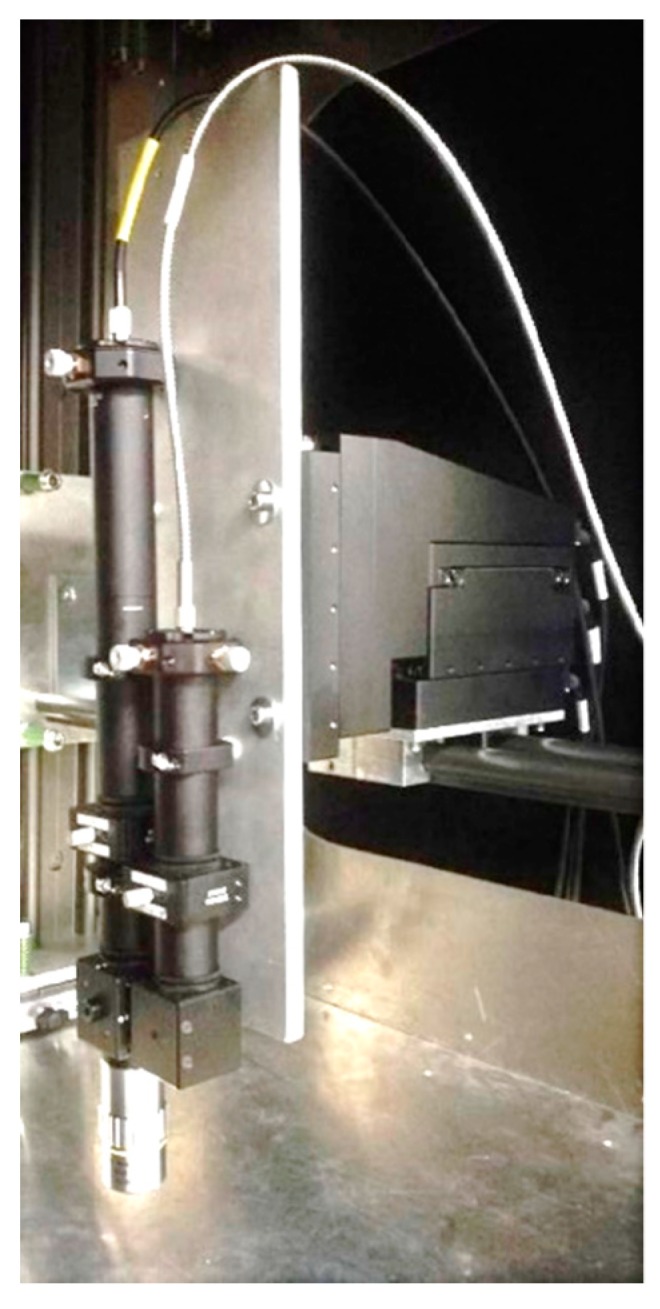
Raman scanning probe with long working distance mounted on a 3D-displaceable table [24].

**Figure 10 micromachines-11-00353-f010:**
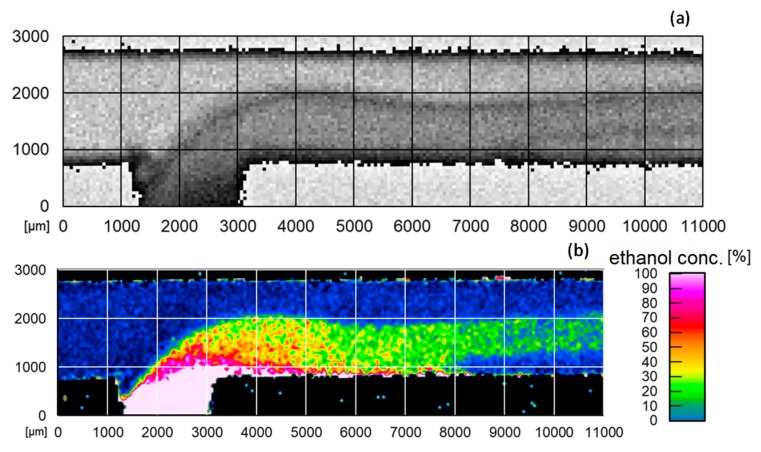
(**a**) 11 × 3 mm Raman scan of ethanol in the reactor and (**b**) 11 × 3 mm false-color image of the ethanol concentration curve in the reactor calculated from quotient method [24].

**Figure 11 micromachines-11-00353-f011:**
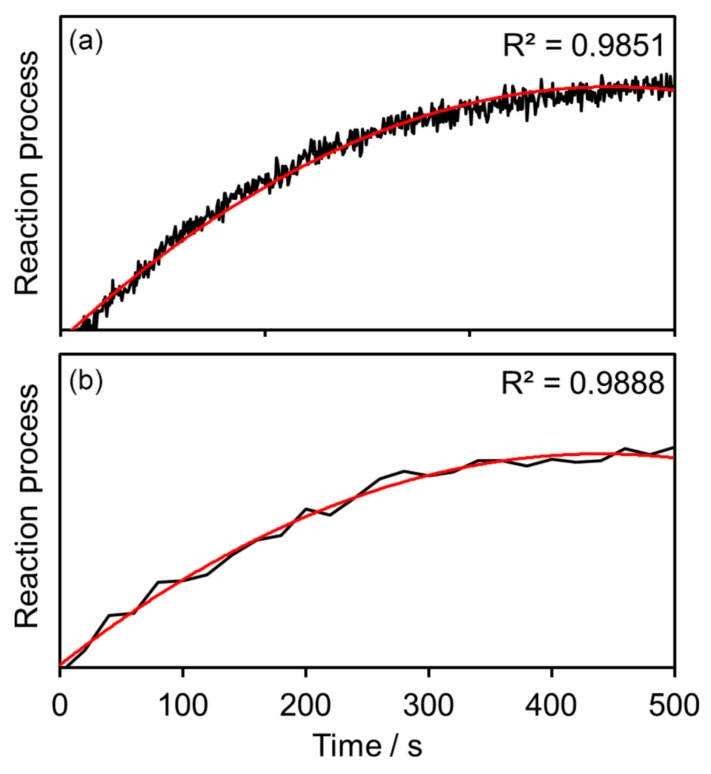
Comparison of (**a**) photometric and (**b**) spectroscopic measurements of CO_2_ binding to amine [18].

**Figure 12 micromachines-11-00353-f012:**
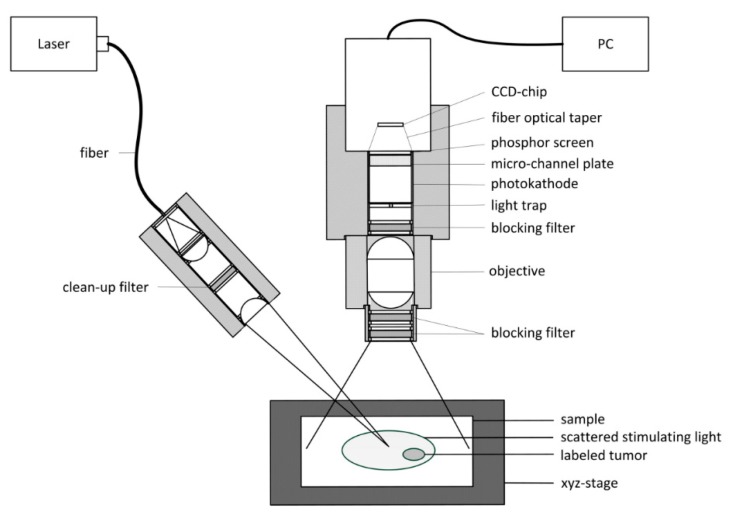
Schematic diagram of a fluorescence imaging system for malignant tissue detection [28].

**Figure 13 micromachines-11-00353-f013:**
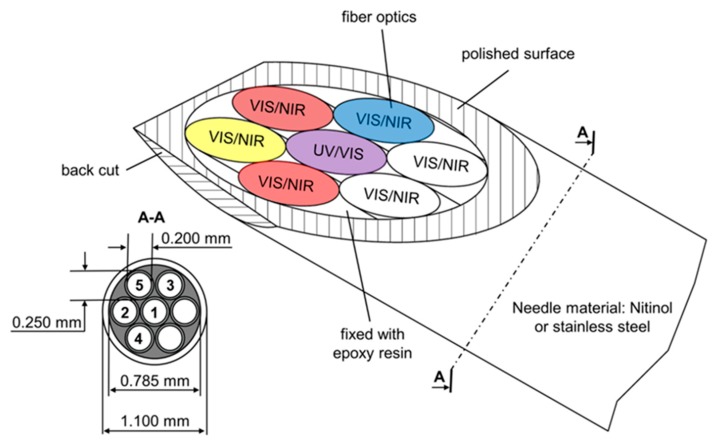
Needle probe for multispectral backscattering measurements. Ultraviolet (UV), visible (VIS), NIR, and fluorescence measurements are possible. The needle is optimized for tissue penetration with a front and back cut [13].

**Table 1 micromachines-11-00353-t001:** Summary of the different measurement techniques.

Technique	Pro	Contra
Ultraviolet/Visible (UV/VIS) Spectroscopy	high sensitivity	low selectivity
NIR (Near-Infrared) Spectroscopy	easily accessible	weak extinction; only measurable at higher concentrations
NIR (Scanning)	high range of detection and monitoring	slow
NIR (Photometric)	fast	smaller wavelength resolution
Mid-Infrared Spectroscopy	high selectivity in combination with a high sensitivity	difficulties with water due to superimposition
Fluorescence Spectroscopy	extremely sensitive	imaging is limited with regard to the selection of the molecule; invasive
Raman Spectroscopy	high selectivity	very weak effect; strong laser or very sensitive detector needed
Particle Detection	very fast	only works for specific particle size

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
