# Peer review of "Molecule Sensitive Optical Imaging and Monitoring Techniques—A Review of Applications in Micro-Process Engineering"

_micromachines, 2020, doi:10.3390/mi11040353_

Round 1

Reviewer 1 Report

A brief summary (one short paragraph) outlining the aim of the paper and its main contributions.

The paper is a review of different optical based techniques for imaging and monitoring molecules in microchannels. The aim is to provide an overview of various techniques. It provides a good introduction of light matter interactions and includes discussion on the advantages and disadvantages of each technique. Its main contribution is a comparison of various methods which should allow researchers to determine which method to use.

Broad comments highlighting areas of strength and weakness. These comments should be specific enough for authors to be able to respond.

Overall the review serves its purpose and provides an adequate overview of each technique.

English language editing would make the manuscript more readable.

The details included for each technique reviewed are generally sufficient but not gratuitous.

The references are all out of order. The first reference in the text should be number 1, followed by 2, 3, 4, etc. Each reference should be referred to at least one in the manuscript. It appears to me that at least reference 1 does not appear. I did not check the others.

Because these techniques are used by collaborative researchers, many of the references come from them. References outside the collaboration group should be used to substantiate each technique.

Additional subsections in Materials and Methods would be helpful, such as Principles, Geometries, Mie Method, Technique Comparison. Consider numbering each technique in the Materials and methods section and organizing them in the same order as in the Results section so that they can be matched up.

Should MIE, as in MIE-theory, appear as Mie instead because it is a name?

A table comparing each technique would help synthesize all the ideas and be an excellent source for readers to refer to when deciding which technique they want to use. The discussion contains most of these ideas, but cannot be quickly looked at like a table. This table could include some of speed/rate, wavelength range, selectivity, depth of field, spatial resolution, temporal resolution, detection limit, cost, etc.

Please locate figures after they have been referred to in the text.

Please use English language labeling in all figures if possible.

Have you obtained permission to reuse the figures from each reference?

Specific comments referring to line numbers, tables or figures. Reviewers need not comment on formatting issues that do not obscure the meaning of the paper, as these will be addressed by editors.

Line 2: I would suggest changing the title to “Molecule Sensitive Optical Imaging and Monitoring Techniques – a Review on Applications in Micro-Process Engineering”. At the very least, it should me molecule, not molecular.

Line 82: The equation should be formatted using mathematical equation formatting, including removal of the star as a multiplication symbol. It is also best considered a part of the text, which would make the sentence directly following it more complete. I would also suggest removing the reference from the equation line.

Line 84: Incomplete sentence. Making the equation part of the text would fix this.

Line 102: Please add discussion about why temperature control is needed in transmission schemes.

Line 110 and 112: It is stated that “this often takes place in UV” then “mostly in the mid-infrared range”. Please make it clear what the distinction is or resolve the contradiction.

Figure 2: is not referred to in the text. Please refer to the figure in the text.

Line 261: Choose a different word than eminent. It does not mean what I believe you mean to say.

Figure 7: Please replace the German labels with English. Please label the color legend to indicate what the colors mean. Please include a scale bar. Please include in the caption what material(s) is(are) contained in the fluid channel (e.g. water and air). Please locate the figure after it is referred to in the text.

Line 294: This small paragraph is best included in line 291 with Figure 8 to follow.

Figure 10: Please locate the figure after it is referred to in the text. Please replace the legend label with English.

Line 314: This paragraph should be appended to the end of the previous paragraph, line 313.

Figure 11: In the caption, the (a) and (b) labels should come before the description, for example “Comparison of (a) photometric and (b) spectroscopic”.

Line 319: Please add discussion about the efficacy of each technique. You show the comparison in figure 11, but fail to discuss what is observed in the figure.

Line 319: It was stated previously that “multichannel-Raman-photometry shines for process control with rather simple matrices” but here you state “complex matrices”. Are the complex matrices found in multichannel-Raman-photometry or elsewhere? This should be clarified.

Line 329: Please state the detection rate. Rather than saying “in the example” use words like “Fiber optic backscattering has been used with a detection rate of __ “ with the reference following, rather that in the middle of the sentence.

Line 336: remove the colon from the section heading.

Line 339: fluorescent marker rather than fluorescence marker.

Line 340 and 341: What is first optical window? Please explain in the text.

Line 345: remove the colon.

Line 352: Unless this refers to scuba divers, the apostrophe should be removed from divers.

Line 364: Hb in an abbreviation for hemoglobin which should first be spelled out before using the abbreviation. Also, it should be deoxy, not desoxy. PCA should also be spelled out before using the acronym.

Author Response

thank you for your time and effort in reviewing our manuscript. We appreciate the thoughtful comments and suggestions that helped to improve the paper. We replied to each annotation to the best of our knowledge and hope to meet your satisfaction.

Yours sincerely,

Matthias Rädle, Julian Deuerling, Marcel Nachtmann

Reviewer 2:

Broad comments highlighting areas of strength and weakness. These comments should be specific enough for authors to be able to respond.

English language editing would make the manuscript more readable.

Some language editing has been carried out.

The references are all out of order. The first reference in the text should be number 1, followed by 2, 3, 4, etc. Each reference should be referred to at least one in the manuscript. It appears to me that at least reference 1 does not appear. I did not check the others.

The references have been sorted according to their appearance and not alphabetical.

Because these techniques are used by collaborative researchers, many of the references come from them. References outside the collaboration group should be used to substantiate each technique.

References outside the collaboration group have been added.

Additional subsections in Materials and Methods would be helpful, such as Principles, Geometries, Mie Method, Technique Comparison. Consider numbering each technique in the Materials and methods section and organizing them in the same order as in the Results section so that they can be matched up.

Subsections have been added.

Should MIE, as in MIE-theory, appear as Mie instead because it is a name?

MIE has been corrected to Mie

A table comparing each technique would help synthesize all the ideas and be an excellent source for readers to refer to when deciding which technique they want to use. The discussion contains most of these ideas, but cannot be quickly looked at like a table. This table could include some of speed/rate, wavelength range, selectivity, depth of field, spatial resolution, temporal resolution, detection limit, cost, etc.

A table for better comparison has been added.

Please locate figures after they have been referred to in the text.

All figures have been relocated.

Please use English language labeling in all figures if possible.

Figure 7 has been correct and now all figures are in english.

Have you obtained permission to reuse the figures from each reference?

Yes and the confirmation has been sent to Mr. Wang from Micromachines Editorial Office.

Specific comments referring to line numbers, tables or figures. Reviewers need not comment on formatting issues that do not obscure the meaning of the paper, as these will be addressed by editors.

All revisions suggested in the specific comments have been carried out.

Reviewer 2 Report

The review paper written by Raedle et al. on optical monitoring techniques is well presented, with fundamental understanding and the currently available techniques well presented.

Generally it is a pleasure to read,  fairly informative and potentially interesting to general readers who would like to know about the field.

However, the authors keep the discussion at an introductory level therefore it would not be extremely technically sophisticated and in-depth. As a review, the reference list with fewer than 25 literature records is not usually enough.

Besides, legends in figure 7 is not written in English which might need to be corrected.

Author Response

thank you for your time and effort in reviewing our manuscript. We appreciate the thoughtful comments and suggestions that helped to improve the paper. We replied to each annotation to the best of our knowledge and hope to meet your satisfaction.

Yours sincerely,

Matthias Rädle, Julian Deuerling, Marcel Nachtmann

Reviewer 1:

Comments and Suggestions for Authors

However, the authors keep the discussion at an introductory level therefore it would not be extremely technically sophisticated and in-depth. As a review, the reference list with fewer than 25 literature records is not usually enough.

Several other references have been added.

Besides, legends in figure 7 is not written in English which might need to be corrected.

Figure 7 has been corrected.